# Core Services: An Introduction to Global Ocean Forecasting

Yann Drillet[1], Matthew Martin[2], Yosuke Fujii[3], Eric Chassignet[4], Stefania Ciliberti[5]

[1]Mercator Ocean International, Toulouse, France
[2]MetOffice, Exeter, UK
[3]Japan Meteorological Agency, Meteorological Research Institute, Tsukuba, Japan
[4]Center for Ocean-Atmospheric Prediction Studies, Florida State University, United States
[5]Nologin Oceanic Weather Systems, Santiago de Compostela, Spain

*Correspondence to*: Yann Drillet (ydrillet@mercator-ocean.fr)

**Abstract.** The capacity in monitoring and forecasting the global ocean is nowadays increased, thanks to the advancements in observing and in modelling the main physical ocean processes and dynamics. This has led to the growth of core services, devoted to providing free and open data, science-driven and based on users' needs. Here we illustrate the fundamental steps that have been developed, over the last decades, for improving the ocean value chain - from access to upstream data like observations to the delivery of products to users for downstream services and applications, with description of worldwide state-of-the-art operational ocean forecasting systems at global scale. We provide also some examples on core services organisation, like the Copernicus Marine Service and many others, which are today available and operating for the provisioning of near real time predictions.

Summary : This article describes the various stages of research and development that have been carried out over the last few decades to produce an operational reference service for global ocean monitoring and forecasting.

## 1 Introduction

Effective monitoring and prediction of the global ocean is nowadays a crucial and demanding need for supporting a wide range of applications– from maritime safety and transports to search and rescue, from offshore industry operations to addressing climate change, including management and planning of fisheries, ecosystems and aquaculture activities. It implies  coordinated actions among marine core services and users through downstream applications as by the "butterfly diagram" shown in Figure 1 from Alvarez-Fanjul et al. (2022): it positions the marine core service as one of the pillars of the whole value chain, in charge of providing high quality information of the ocean state by combining observations and numerical modeling, timely delivered to users for the implementation of tailored tools for decision making.

Therefore, a core service should have by definition the following characteristics:

- It feeds from ocean observations, both from satellite and in-situ sources.

- It provides reliable access (production requirements are defined and information provided to users on target delivery time, timeliness and monitoring of dedicated Key Performance Indicators (KPIs)) to both quality controlled measured and forecasted ocean data.
- It is user-driven, and specific support to the users is provided.
- It generates data useful for final and intermediate users, enabling the latter to produce tailored information for final users.
- The development, evolution and operations are done under a well-controlled planning, ensuring availability, timeliness and quality of the resulting products.

The concept of a "core service" was developed in the framework of the Copernicus program, but the idea of providing reliable and up-to-date information on the state of the environment is universal. There are other global ocean services, such as the Global Ocean Observing System (GOOS), which also provide information on the state of the world's oceans and seas. However, the specific services offered and the way in which they are organized may differ between programs, so not all of them can be considered "core services" in the sense developed by Copernicus. Here we define a core service as the provision of open and free data together with dedicated user support with the characteristics described in the next section.

In this chapter, we will focus on the general characteristics of existing Global Ocean Forecasting Systems, their collocation in the framework of marine core services and existing international initiatives that support scientific networking and activities for improving and advancing numerical ocean predictions.

## 2 Global Ocean Forecasting Systems: where we are today

Last decade has been characterized by vibrant advancements in numerical ocean modeling and observational networks that opened new opportunities for improving global ocean monitoring and forecasting. The last review on the status of ocean forecasting systems described in Tonani et al. (2015) outlined that 12 global systems were regularly operating up to 2015 across the world – from France, UK, Norway and Italy to USA, Canada and Brazil, from Australia to Japan and China to India – with an increase of 30% with respect to 2009 where only 7 were providing forecast products. These actions were and still are supported by an international coordinated effort promoted by the Global Ocean Data Assimilation Experiment (GODAE) over 3 main steps:

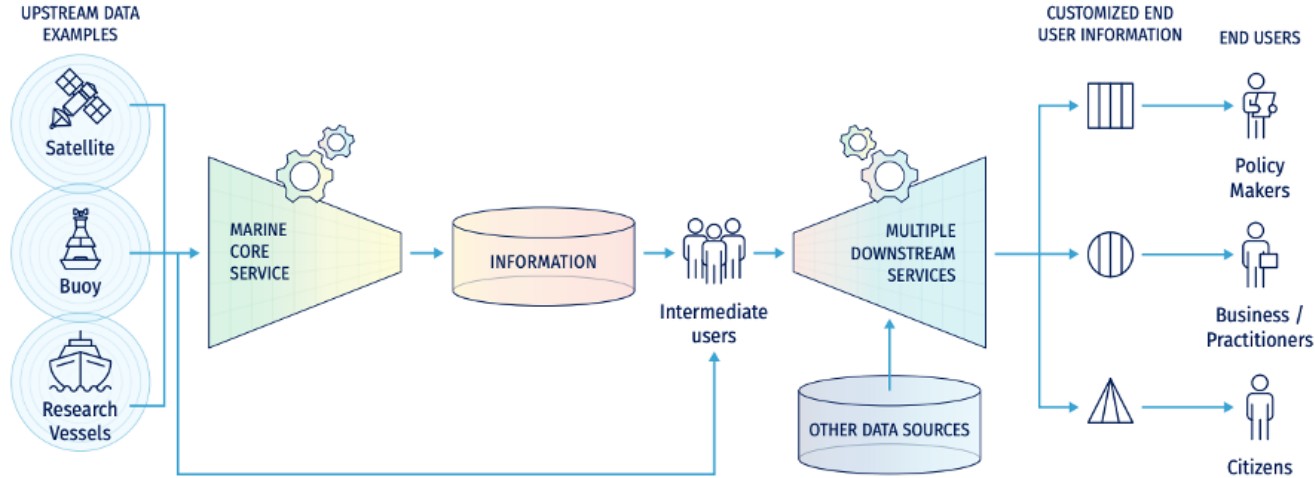

**Figure 1: The ocean value chain (from Alvarez-Fanjul et al., 2022)**

- Phase 1: The Experiment (Bell et al., 2009). GODAE started in 1998 and developed over 10-years, with the main scopes of
  - Applying state-of-the-art ocean models and data assimilation methods for producing short term forecast and for providing initial/boundary conditions for regional-to-coastal subsystems.
  - Providing global ocean analysis to understand the ocean state, to improve predictability and to support the design and the effectiveness of the global ocean observing system.
- Phase 2: "Science to underpin Societal Needs" (Bell et al., 2015; Schiller et al., 2015). Following the first step, over the next 10 years, GODAE OceanView consolidated the coordination by launching new activities devoted to developing predictive systems to meet users' needs. Such activities included
  - The consolidation and improvement of global (and regional) systems.
  - The scientific evolution for the next generation of systems.
  - The exploitation of this capacity in other contexts, like ocean reanalysis, weather forecasting, seasonal and decadal prediction, climate change, coastal impacts).
  - The assessment and the design of the ocean observing network.
- Phase 3: Advancing the science of ocean prediction: OceanPredict. Since 2019, GODAE OceanView became OceanPredict, with the main scope to enhance ocean prediction within an overall operational oceanography context (OceanPredict Strategy 2021-2030, 2021), by working on 5 major drivers[1]:
  - Data Assimilation: to improve ocean forecasting, it is also needed to improve data assimilation capacity.

---

[1] https://oceanpredict.org/about/strategy/goals/

- Verification: monitoring and demonstrating improved accuracy and utility of ocean analysis and forecasting products resulting from OceanPredict contributions, by coordinating regular system intercomparisons and verifications.
- Observing system evaluations: contributing to projects and assessment to better determine observation impact and feeding back to the observing system community information about opportunities for further improving the impact of observations on forecasting skill.
- Models: collaborating with R&D groups through OceanPredict task teams to improve ocean predictions in shelf seas and coastal environment, for biogeochemical variables and for coupled environmental prediction systems.
- Visualization: collaborating with ocean product developers and ocean services to improve visualization and accessibility tools for predictions and observations.

GODAE/OceanPredict Science Team, that includes more than 30 experts that are leaders in the field of operational oceanography from national, international and intergovernmental organizations, is in charge of maintaining updated information about the current global ocean forecasting capacity of the physical and biogeochemical components, including technical description of the systems and available viewing services. In Alvarez-Fanjul et al. (2022), detailed complementary inventories of global ocean systems available worldwide are given. Table 1 summarizes services provided by the operational centers (technical characteristics of operational ocean forecasting systems are given in Alvarez-Fanjul et al. (2022), showing that state of the art ocean model and data assimilation methods are used to produce standard products including main Essential Ocean Variables (EOVs)).

**Table 1: Updates on the inventories as given in Alvarez-Fanjul et al. (2022) and by the OceanPredict[2], with focus on provided essential ocean variables (EOV) and summary of offered service.**

| System | EOV | Service |
|---|---|---|
| GIOPS (Global Ice Ocean Prediction System) 1/4° resolution | Temperature, salinity, sea surface height, zonal and meridional velocity components, sea ice concentration, sea ice thickness, northward sea ice velocity. | Daily means and 3h average surface fields. From 2014 to present https://science.gc.ca/site/science/en/concepts/prediction-systems/global-ice-ocean-prediction-system-giops |
| ESSO-INCOIS (Earth System Science Organisation-Indian | Temperature, salinity, sea surface height, zonal and meridional | 6h average hourly fields for 5 days forecast https://incois.gov.in/ |

---

[2] https://oceanpredict.org/science/operational-ocean-forecasting-systems/ocean-products-services/

| | velocity components, mixed layer depth. | |
|---|---|---|
| National Centre for Ocean Information Services) 1/4° resolution | velocity components, mixed layer depth. | |
| MOVE (Multivariate Ocean Variational Estimation) nested grid from **1**° at global scale to **1/33**° in Japanese Seas | Temperature, salinity, sea surface height, zonal and meridional velocity components, sea ice concentration. Daily mean | Daily mean from October 2020 to present, 31 days forecast for the North Pacific, and 11 days forecast with a higher resolution for the Japan area. https://www.jmbsc.or.jp/jp/online/file/f-online23100.html (in Japanese) |
| OceanMAPS (Ocean Modelling and Analysis Prediction System) 1/10° resolution | Temperature, salinity, sea surface height, zonal and meridional velocity components. | Daily means. From 2007 to present https://research.csiro.au/bluelink/global/forecast/ |
| GLOMFC (Global Monitoring Forecating Center from Copernicus Marine Service) **1/12**° resolution | Temperature, salinity, sea surface height, zonal and meridional velocity components, mixed layer depth, bottom temperature. Chlorophyll, dissolved inorganic carbon in sea water, iron, oxygen, nitrate, phosphate, silicate, primary production, alkalinity, pH, surface partial pressure of carbon dioxide in sea water, | Hourly, daily and monthly means since 2020 to + 10 days for the physical component; daily and monthly means from 2021 to present for the biogeochemical component; hourly instantaneous from 2021 to + 10 days for the wave component https://marine.copernicus.eu/about/producers/glo-mfc |

| | volume attenuation coefficient of downwelling radiative flux. Significant wave height, wave period, peak period, wave direction, wave maximum height, Stokes drifts, swell significant heights, swell wave directions. | |
|---|---|---|
| FOAM (Forecast Ocean Assimilation Model) 1/12° resolution | Temperature, salinity, sea surface height, zonal and meridional velocity components, sea ice concentration, sea ice thickness, sea ice velocity. | Daily forecasts out to 7 days producing data with daily and 3h frequency. https://www.metoffice.gov.uk/research/weather/ocean-forecasting/ocean-development |
| GOFS3.1 (Global Ocean Forecasting System) 1/12° resolution | Temperature, bottom temperature, salinity, sea surface height, zonal and meridional velocity components, sea ice concentration, sea ice thickness, sea ice velocity. | 3h means. From 2018 to + 4 days https://www.hycom.org/dataserver/gofs-3pt1/analysis |
| GOFS16 (Global Ocean Forecasting System) 1/16° resolution | Temperature, salinity, sea surface height, | Daily means of + 5 days https://gofs.cmcc.it/ |

| | zonal and meridional velocity components. | |
|---|---|---|
| NMEFC (National Marine Environmental Forecasting Center) 1/12° resolution | Temperature, salinity, velocities, sea ice. | Daily means and 5-day forecast. https://www.nmefc.cn/ybfw/seacurrent/Global |

**3 The Copernicus Marine Service as reference core service and its offer for the global ocean**

In the framework of the EU Copernicus program, the Copernicus Marine Service is organized to provide operational service to external users and to get user feedback to improve an user-driven service. It has been defined with the following specificities:

- Free access to reliable up-to-date and historic data is a key for enhanced knowledge and better understanding of our oceans.
- Copernicus Marine Service provides data from satellites, in-situ sensors, and numerical models covering the Global
Ocean and the European Regional Seas and associated product quality information.
- Information on past, present and future trends are made available to empower all users who want to drive the Blue Economy, scientific innovation and support sustainable ocean initiatives.
- Anyone can use the data: scientists, policy makers, entrepreneurs and ordinary citizens, from all over the world.
- Tailored services and training, adapted to different levels of expertise and familiarity with ocean data.
- Users can get help from the Copernicus Marine User support team.
- Interoperability between different producers is ensured between all the products available in the catalog and to allow connection between the producers.
- Standards (including best practices) are defined and applied by the producers for the products (resolution, frequency, variable, time series, forecast length …), the format, the quality information and the timeliness.

User feedback is organized within the core service by collecting and analyzing information on access to data, the services offered and user support through surveys and training sessions, as well as through a user uptake programme in the form of projects and thanks to a group of experts (i.e., the Champion User Advisory Group) which analyzes and summarizes needs. Access to the Copernicus Marine Product Catalog is done through https://marine.copernicus.eu/ .

Copernicus Marine Service is organized around Thematic Assembly Centers (TAC) and Monitoring Forecasting Centers
(MFC) (Figure 2). TACs process data acquired from satellite ground segments and in situ platforms to produce real-time (today) and reprocessed (30 years historic) products. They are organized by thematic hubs including sea ice, wind, sea level, in situ, ocean color, sea surface temperature, wave and multi observations. MFCs run ocean numerical models assimilating data provided by TAC data to generate reanalysis (30 years in the past), analysis (today) and 10-day forecasts of the ocean.

They are organized in geographical areas, including global ocean and European Seas such as Arctic ocean, Baltic Sea, Atlantic
European NorthWest Shelves, Iberian-Biscay-Irish Seas, Mediterranean Sea and Black Sea.

Focusing on global ocean forecasting systems, the Copernicus Marine, through the GLO-MFC, provides marine data (waves, currents, temperature, salinity, sea level and biogeochemistry) for the world's oceans, Atlantic, Indian, Pacific, Arctic and Antarctic, and the European seas. Past, present and future are covered by this data, providing information for 30 years in the past up to 10 days in the future. The portfolio of products (as summarized also in Table 1 ~~Table 2.1-1~~) includes:

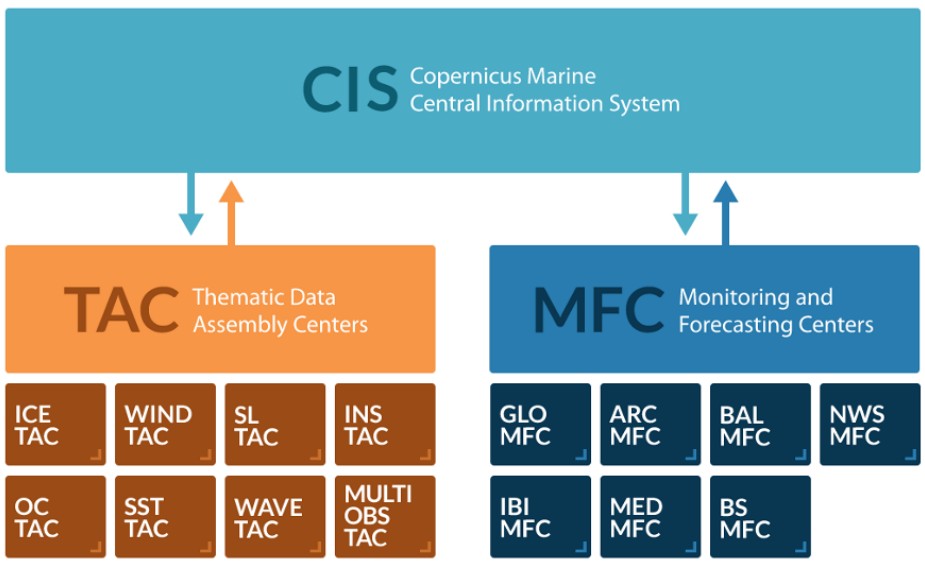

**Figure 2: Organisation of Copernicus Marine Service including Thematic Assembly Centers (TAC), that provide ocean observations, and Monitoring and Forecasting Centers (MFC), that provide reanalysis and forecast at global scale and for the European Seas. Exchanges with users are done through the Central Information System (CIS).**

- **Near Real Time (NRT)** datasets for physics and waves at 1/12° resolution and for biogeochemistry at 1/4°, forced by ECMWF IFS atmospheric forecasting product (see https://www.ecmwf.int/en/elibrary/81235-evaluation-ecmwf-forecasts-including-2021-upgrade for a description of the systems and their evolutions) .
  - o Global Ocean Physics Analysis and Forecast, run by Mercator Ocean International, provides analysis and forecast of the 3D ocean regularly every day. The timeseries is aggregated in time to reach a 2 full years' time in a sliding window to + 10 days. The core model is based on NEMO (Nucleus for European Modelling of the Ocean) ( v3.6 coupled to LIM3 sea ice model: it assimilates temperature and salinity profiles as well as sea surface temperature, sea ice concentration and sea level anomaly provided by corresponding TACs using SAM2 data assimilation scheme. Details in Le Galloudec et al. (2023) and Lellouche et al. (2023). An example of the sea surface temperature forecast field is given in Figure 3.
  - o Global Ocean Biogeochemistry Analysis and Forecast, run by Mercator Ocean International, provides analysis and forecasts of the 3D global ocean updated weekly. The timeseries is aggregated similarly to the physical system. The core model is based on NEMO v3.6 online coupled to PISCES for the biogeochemical

component: it assimilates satellite ocean color provided by OC-TAC using SEEK (Singular Evolutive Extended Kalman) data assimilation scheme. Details in (Lamouroux et al., 2023a and 2023b)

- o Global Ocean Waves Analysis and Forecast, run by MeteoFrance, provides analysis and forecasts of the global ocean sea surface waves. The core model is MFWAM, with spectral resolution of 24 directions and 30 frequencies: it uses optimal interpolation for the assimilation of significant wave height from altimeters. Details are given in Dalphinet et al. (2023) and Aouf et al. (2023).

- **The Multi-Year (MY)** datasets for physics at 1/12° resolution, for biogeochemistry at 1/4°, and for waves at 1/5°, forced by ECMWF ERA5 atmospheric reanalysis (Hersbach et al, 2020).

  - o Global Ocean Physics Reanalysis, run by Mercator Ocean International, provides reanalysis of the global ocean covering the altimetry period (from 1993 onward). The core model is based on NEMO v3.1, coupled to LIM2 (Louvain ice Model) ice model and implementing SAM2 (System assimilation Mercator) scheme for the assimilation of reprocessed observations such as satellite sea surface temperature, sea ice concentration and sea level anomaly, in situ temperature and salinity profiles. Details are given in Drevillon et al. (2023a, 2023b).

  - o Global Ocean Biogeochemistry Hindcast, run by Mercator Ocean International, provides biogeochemical hindcasts for the global ocean over a period starting in 1993. The core model is based on NEMO v3.6 coupled to PISCES. Details are given in Le Galloudec et al. (2022) and Perruche et al. (2019).

  - o Global Ocean Waves Reanalysis, run by Mercator Ocean International, provides the global wave reanalysis since 1993. The core model is MFWAM coupled to an optimal interpolation scheme for the assimilation of significant wave height provided by altimeters. Details are given in Law-Chune et al. (2023a, 2023b).

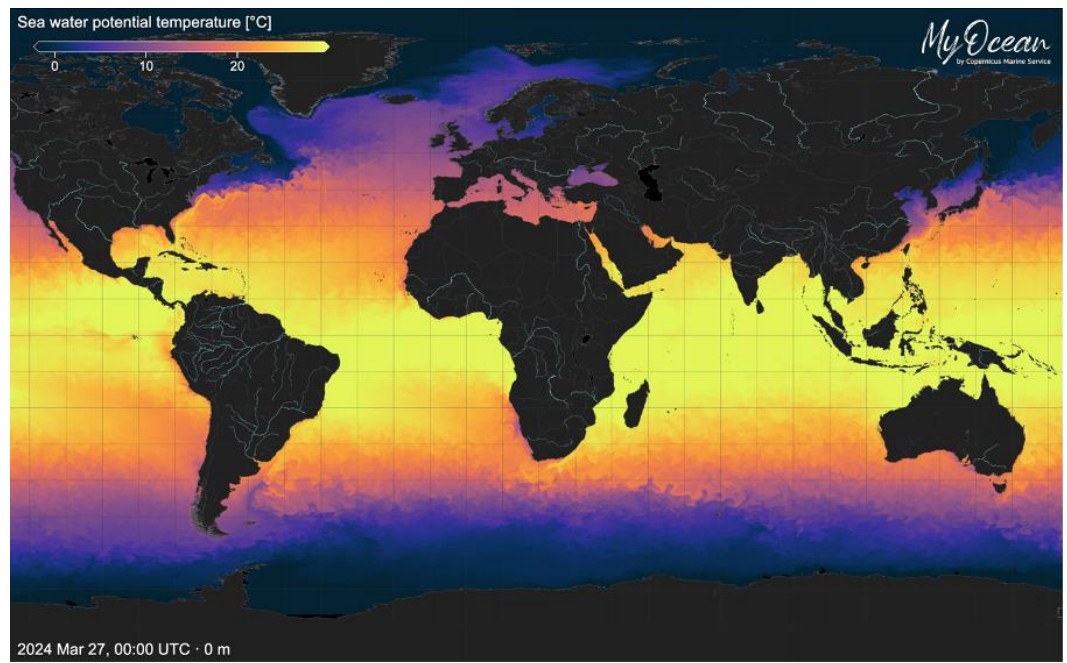

**Figure 3: Sea surface temperature as predicted by the Global Ocean Physical Analysis and Forecasting System on the 27 March 2024: visualization provided by the Copernicus Marine Service - MyOceanPro Viewer.**

## 4 Other worldwide Ocean Services

The list of operational oceanography centres and associated services is evolving rapidly, and the centralisation and updating of this information is one of the important activities for international coordination and is carried out within the framework of ocean predict (https://oceanpredict.org/science/operational-ocean-forecasting-systems/ocean-products-services/) and the ocean prediction decade collaborative centre where a dedicated atlas is provided (https://www.unoceanprediction.org/en/atlas/people?lat=16.46769474828897&lng=23.5546875&zoom=2).

- The National Oceanic and Atmospheric Administration (NOAA) is the reference agency in the US that provides understanding and predictions of changes occurring in climate, weather, ocean and coasts, sharing knowledge and information and conserving and managing coastal and marine ecosystems and resources. The NOAA's National Ocean Service operates with the Centre for Operational Oceanographic Products and Services (CO-OPS) for gathering accurate, reliable, and timely water-level and current measurements. The NOAA's National Weather Service[3] provides, through the Environmental Modeling Center, the Global Real-Time Ocean Forecast System products[4], delivered via FTP. Visualization of nowcast/forecast products and reference metrics are provided as well through a dedicated webpage available at https://polar.ncep.noaa.gov/global/.

---

[3] https://oceanservice.noaa.gov/
[4] https://polar.ncep.noaa.gov/global/

- From the collaboration between Environment and Climate Change Canada, Fisheries and Oceans Canada, and National Defence departments, the Government of Canada supports the Canadian Operational Network of Coupled Environmental PredicTion Systems (CONCEPTS[5]) for the monitoring of the met-oceanographic conditions in the Country. CONCEPTS provides operational access to real-time forecasts through dedicated web services (i.e., geospatial web services and third-party websites) including bulletins produced with static images. CONCEPTS includes prediction systems like the Global Ice Ocean Prediction System (GIOPS) with delivery of 10 days forecast of daily ocean and sea ice analysis together with the regional systems (e.g., the Regional Ice Ocean Prediction System RIOPS and the Regional Deterministic Prediction System Coupled over the Gulf of St. Lawrence RDPS-CGSL) and a dedicated one for the Great Lakes (i.e., the Water Cycle Prediction System Coupled over the Great Lakes WCPS-CGL[6]).

- The European Centre for Medium-Range Weather Forecasts (ECMWF) develops and maintains and operational system called OCEAN5[7], a global eddy-permitting ocean-sea ice ensemble with 5 members from 1979 to present. It includes a Behind-Real-Time (BRT) component to produce ocean reanalysis from 1979 to present (ORAS5[8]) and a Real-Time (RT) component, initialized from the last BRT analysis to compute an analysis up to real time every day using a variable assimilation window. Data are accessible through the Copernicus Climate Data Store and are used for performing past reconstruction of the ocean climate state at global scale.

- The Australian Government Bureau of Meteorology is Australia's national weather, climate and water agency[9]. It provides Marine and Ocean products such as wind maps, tide predictions, sea temperature and currents, wave maps, seasonal ocean temperature. The service is for citizens and society, so the communication is done through VHF, radio and radiofax, internet and through satellite.

- The Japan Meteorological Agency (JMA[10]) is the reference Japanese agency for monitoring weather, earthquakes and volcanoes activities. The ocean component of the JMA carries out oceanographic and marine meteorological observations in the western North Pacific and seas adjacent to Japan. Additionally, it operates with a set of operational ocean data assimilation and prediction systems named MOVE for coastal disaster prevention, supporting fishery, marine transportation, and marine industry, and providing the oceanic initial conditions for the coupled atmosphere-ocean forecasting systems(Hirose et al., 2019; Fujii et al., 2023; Yamanaka et al., 2023).

- .

---

[5] https://science.gc.ca/site/science/en/concepts
[6] https://science.gc.ca/site/science/en/concepts/prediction-systems
[7] https://www.ecmwf.int/en/research/climate-reanalysis/ocean-reanalysis
[8] https://cds.climate.copernicus.eu/cdsapp#!/dataset/reanalysis-oras5?tab=overview
[9] http://www.bom.gov.au/?ref=hdr
[10] https://www.jma.go.jp/jma/indexe.html

- The China Meteorological Administration (CMA[11]) is an operator, service-provider, and regulator in weather forecasting and warning, climate prediction and public meteorological services. The National Meteorological Centre (NMC) undertakes the responsibility of issuing forecasts and warnings for 13 different types of hazardous weather conditions within the next 24 hours. These include typhoons, heavy rain, severe convective weather, blizzards, cold waves, gales at sea, sandstorms, low temperatures, high temperatures, frosts, ice storms, heavy fog, and haze. The Beijing Climate Centre operates its own global ocean system for the monitoring of the ocean climate events like El Nino in the Central and Eastern Equatorial Pacific[12].

- The National Marine Environmental Forecasting Center (NMEFC[13]) is the national operation and research center for marine environmental forecasting, marine hazard warning, and provides advisory information for public policy, decision making, and the socio economic and sustainable development, which is a public commonweal institution directly under Ministry Natural Resources of China.

- Mercator Ocean International (MOi[14]) is a non-profit organisation, in the process of transforming into an intergovernmental organisation, providing ocean science-based services of general interest focused on the conservation and the sustainable use of the oceans, seas and marine resources. After running the European MyOcean projects since 2009, Mercator Ocean was officially appointed by the European Commission on November 11th, 2014 to implement the European ocean-monitoring service, the Copernicus Marine Service, as part of the European Earth observation programme, Copernicus.

- The Met Office[15] is the UK's national weather and climate service and produces operational global and regional ocean forecasts on a daily basis using the FOAM system as well as waves, storm surge and ecosystem predictions. Research effort is reinforced by a close collaboration with academic groups, including those in the National Partnership for Ocean Prediction (NPOP).

- The CMCC[16] Foundation (Euro-Mediterranean Center on Climate Change) is an international, independent, multi-disciplinary research center that studies the interaction between climate change and society. They produce advanced climate research developing cross-cutting and multidisciplinary analyses and data that combine first-class climate modeling with climate change impact modeling and environmental economics.

- ESSO-INCOIS[17] was established as an autonomous body in 1999 under the Ministry of Earth Sciences (MoES) and is a unit of the Earth System Science Organisation (ESSO). ESSO- INCOIS is mandated to provide the best possible

[11] https://www.cma.gov.cn/en/
[12] https://www.cma.gov.cn/en/forecast/news/202402/t20240229_6093860.html
[13] http://www.nmefc.cn/hailiu/quanqiu.aspx
[14] https://www.mercator-ocean.eu/
[15] https://www.metoffice.gov.uk/
[16] https://www.cmcc.it/
[17] https://incois.gov.in/portal/aboutus

ocean information and advisory services to society, industry, government agencies and the scientific community through sustained ocean observations and constant improvements through systematic and focussed research.

•

## 5 Conclusion

The development of operational ocean analysis and forecasting systems began in the late 1990s for institutional and expert users. The first systems produced analyses and forecasts of the physical ocean at intermediate resolutions (between 1° and ¼°) and frequencies that were daily at best. The output from these systems was made available directly on supercomputers or on

archive centres or ftp servers. The progress made in production systems was accompanied by progress in dissemination systems, visualisation tools, data processing and in the support provided to users in order to create what are currently called core services. The horizontal resolution of global models now reaches a few kilometres, and the temporal resolution of forecasts updated daily can be hourly, with assimilated data and model forcings also having progressed in line with the targeted resolutions. Data is now distributed on cloud servers in optimised formats, enabling large volumes of data to be viewed and

handled efficiently. Standardisation of the associated documentation, monitoring of operational production and user support mean that these operational products can be used more easily. The number of users of operational oceanography products has risen sharply, with some core services currently able to serve several tens of thousands of users. Digital Twin's developments for the ocean will make it possible to integrate new technologies and, in the near future, will represent an important evolution in the core service for operational oceanography.

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

**Competing interests**

The contact author has declared that none of the authors has any competing interests.

**Data and/or code availability**

This can also be included at a later stage, so no problem to define it for the first submission.

**Authors contribution**

This can also be included at a later stage, so no problem to define it for the first submission.

**Acknowledgements**

**Authors acknowledge European Commission and Copernicus Marine Service, Oceanpredict and the decade collaborative center OceanPrediction in the framework of the UN decade of Ocean Science for sustainable Development.**