# Peer review of "Core Services: An Introduction to Global Ocean Forecasting"

_State of the Planet, 2024_

## Author Response (AR1)

REVIEW sp-2024-38

**Core Services: An Introduction to Global Ocean Forecasting**

Yann Drillet, Matthew Martin, Yosuke Fujii, , Eric Chassignet, and Stefania Ciliberti

REVIEW 1

General comments:

The contents of the paper are an appropriate and well-written introduction to global ocean forecasting and associated services.

The paper is lacking a summary and/or conclusion section. I suggest briefly summarizing the current state of the art and where core services might be 20 years from now.

Added section 5 Conclusion

The development of operational ocean analysis and forecasting systems began in the late 1990s for institutional and expert users. The first systems produced analyses and forecasts of the physical ocean at intermediate resolutions (between 1° and ¼°) and frequencies that were daily at best. The output from these systems was made available directly on supercomputers or on archive centres or ftp servers. The progress made in production systems was accompanied by progress in dissemination systems, visualisation tools, data processing and in the support provided to users in order to create what are currently called core services. The horizontal resolution of global models now reaches a few kilometres, and the temporal resolution of forecasts updated daily can be hourly, with assimilated data and model forcings also having progressed in line with the targeted resolutions. Data is now distributed on cloud servers in optimised formats, enabling large volumes of data to be viewed and handled efficiently. Standardisation of the associated documentation, monitoring of operational production and user support mean that these operational products can be used more easily.  The number of users of operational oceanography products has risen sharply, with some core services currently able to serve several tens of thousands of users. Digital Twin's developments for the ocean will make it possible to integrate new technologies and, in the near future, will represent an important evolution in the core service for operational oceanography.

Specific comments:

Page 1, line 33: please define what you mean by "reliable access" to core services, e.g. in terms of latency, quality etc.

Added in the text : requirements to producers and information provided to users on target delivery time, timeliness and monitoring of dedicated KPIs

Page 2, line 35: ""enabling the latter to produce…".

Text modified

Page 3, line 69-70: what about also exploiting the benefits from ocean reanalyses?

Text modified

> o The exploitation of this capacity in other contexts, like ocean reanalysis, weather forecasting, seasonal and decadal prediction, climate change, coastal impacts).

Page 4, line 80: I suggest adding "… and feeding back to the observing system community information about opportunities for further improving the impact of observations on forecasting skill" or similar.

Text modified

Observing system evaluations: contributing to projects and assessment to better determine observation impact and feeding back to the observing system community information about opportunities for further improving the impact of observations on forecasting skill

Page 4, line 88: What about biogeochemical forecasting component in addition to the physical component)?

Text modified

about the current global ocean forecasting capacity of the physical and biogeochemical components

Page 4, Table 1: Can the authors confirm that the Global MFC operated by the Copernicus Marine Service is currently the only centre which produces operational forecasts of bgc variables? Please update table if other forecasting centres also produce forecasts of bgc variables.

Currently, the Global MFC is the the only operational services provided global biogeochemical forecast.

Page 7, line 98: Again, what is the exact definition of "reliable", e.g. is it 24/7 services, quality-controlled data delivery, low latency of forecasting products etc.?

Added in text

Page 8, line 125: Where is Table 2.1-1? Should this be Figure 2?

Corrected Table 1 instead of table 2

Page 8, line 131: ECMWF IFS atmospheric forecasting product: please add reference.

Reference to website documentation added in the text

Page 9, line 149: ECMWF ERA5 atmospheric reanalysis: please add reference.

Added in the text (Hersbach et al, 2020).

Technical corrections:

Page 1, lines 26-27: "... It implies a coordinated actions among marine core services ...". Delete "a".

Page 2, line 35: Replace "being the latter able" with "enabling the latter to produce...".

REVIEW 2

1. Table 1 refer to the article by Alvarez-Fanjul et.al (2022) and cited many operational systems from this article. However, in my opinion, it is not covered enough. I suggest to list all systems which metioned by the OceanPredict website.

The list of systems identified in oceanpredict is much broader than the system identified here that should answer to global core service definition. It's why there is only a selection of service identified in table 1.

2. Can all abbreviations be listed at the end of the article?
   Abbreviations are defined in the text.
   This is the list of abbreviations used, I suggest not to include a long list of abbreviations in this short and synthetic article.

CIS : Central Information System
ECMWF : European Centre for Medium-Range Weather Forecasts
EOVs : Essential Ocean Variables
ESSO-INCOIS : Earth System Science Organisation-Indian National Centre for Ocean Information Services
FOAM : Forecast Ocean Assimilation Model
GIOPS Global Ice Ocean Prediction System
GLOMFC : Global Monitoring and Forecasting Center
GOFS Global Ocean Forecasting System
GOOS : Global Ocean Observing System
GODAE : Global Ocean Data Assimilation Experiment
KPIs : Key Parameter Indicators
LIM : Louvain Ice Model
MOi : Mercator Ocean Internatinal
MOVE : Multivariate Ocean Variational Estimation
MY : Multi year
NEMO : Nucleus for European Modelling of the Ocean
NMEFC : National Marine Environmental Forecasting Center
NOAA : National Oceanic and Atmospheric Administration
NRT : Near Real Time
OceanMAPS : Ocean Modelling and Analysis Prediction System
SAM : System Assimilation Mercator
SEEK : Singular Evolutive Extended Kalman
TAC : Thematic Assembly Centers

3. Line 125 metioned the portifolio of summermaried in Table 2.1. However, I have not found table 2. Please check it.
Corrected reference to table 1 not table 2.

4. Chinese ocean services provided by NMEFC not CMA. Please check it.

NMEFC operational global system added in added in the table and in the list of worldwide ocean services